

# Clutter Mitigation, Multiple Peaks, and High-Order Spectral Moments in 35-GHz Vertically Pointing Radar Velocity Spectra

Christopher R. Williams[1], Maximilian Maahn[2,3], Joseph C. Hardin[4], Gijs de Boer[2,3]

[1] Ann and H.J. Smead Aerospace Engineering Sciences Department, University of Colorado Boulder, Boulder, CO, 80309, United States
[2] Cooperative Institute for Research in Environmental Science (CIRES), University of Colorado Boulder, Boulder, CO, 80309, United States
[3] NOAA Earth System Research Laboratory (ESRL), Physical Sciences Division, Boulder, CO, 80305, United States
[4] Pacific Northwest National Laboratory (PNNL), Richland, WA, 99354, United States

*Correspondence to*: Christopher R. Williams (Christopher.Williams@colorado.edu)

**Abstract.** This study presents three separate processing methods to improve high-order moments estimated from 35-GHz (Ka-band) vertically pointing radar Doppler velocity spectra. The first method removes Doppler shifted ground clutter from spectra collected by a US Department of Energy (DOE) Atmospheric Radiation Measurement (ARM) program Ka-band zenith pointing radar (KAZR) deployed at Oliktok Point, Alaska. Multiple pathways through antenna side-lobes and reflections off a rotating scanning radar antenna located 2 m away from KAZR caused Doppler shifts in ground clutter returns from stationary targets 2.5 km away. After removing clutter in the recorded velocity spectra, the second processing method identifies multiple atmospheric peaks in the spectra and estimates high-order moments for each unique peak. Multiple peaks and high-order moments were estimated for both original 2-s and 15-s averaged spectra. The third processing method improves the spectrum breadth, skewness, and kurtosis estimates by removing 2-s velocity variability during 15-s averaging intervals. Assuming the cloud and precipitation microphysical properties do not change during the 15-s interval, shifting individual 2-s spectra to a common 15-s mean velocity before averaging removes 2-s temporal scale turbulent broadening.

Consistent with previous studies, this work found that spectrum skewness assuming only a single spectral peak was a good indicator of two hydrometeor populations (for example, cloud and drizzle particles) being present in the radar pulse volume. Yet, after dividing the spectrum into multiple peaks, velocity spectrum skewness for individual peaks is near zero, indicating nearly symmetric peaks. This suggests that future studies should use velocity skewness of single peak spectra as an indicator of possible multiple peaks and then use multiple-peak moments for quantitative studies. {267 words}



## 1 Introduction

Vertically pointing radars operating in the Ka-band (35 GHz) are important remote sensing instruments providing quantitative and high-resolution observations for studying the vertical structure and dynamics of clouds and precipitation (Görsdorf et al., 2015). Vertically pointing radars increase their sensitivity by transmitting multiple pulses and produce Doppler velocity spectra for each range gate and dwell. The temporal evolution and vertical structure of these spectra contain microphysical and dynamical cloud and precipitation information.

By using narrow beamwidth antennas to reduce spectrum broadening due to sub-pulse volume turbulence, the recorded spectra are often non-Gaussian shaped and contain multiple peaks due to the presence of different particle size distributions within the radar pulse volume (Kollias et al., 2016). Under certain atmospheric conditions, mixed phase clouds occur and contain both liquid and ice phase particles within the same radar pulse volume (Shupe et al., 2004; Kalesse et al., 2016). Thus, the number of spectrum peaks and their shape provide microphysical information of the particle size distributions. Estimating higher-order spectral moments, including velocity spectrum skewness and kurtosis, extracts microphysical information from the full Doppler spectrum (Luke and Kollias, 2013). These high-order moments are inputs to time-height analyses exploring microphysical and dynamical cloud processes (Maahn and Löhnert, 2017). One caveat for this analysis paradigm is the need for clean radar Doppler velocity spectra void of non-atmospheric signals, including ground clutter. Thus, pre-processing and cleaning of Doppler spectra are often needed before microphysical and dynamical information can be extracted from vertically pointing cloud radar observations.

This study presents three separate methods to improve high-order moments estimated from Doppler spectra. First, Doppler velocity spectra are cleaned by removing ground clutter. Second, multiple peaks are identified within the Doppler spectra. Finally, spectrum skewness estimates are improved by removing turbulent broadening effects at the 15-s scale.

Ground clutter in scanning and vertically pointing radar observations is a pervasive problem (Sato and Woodman, 1982). Ground structures (including buildings, trees, and power lines) act as hard targets reflecting radar waves back to the radar. Since these ground structures are stationary, except for oscillatory trees and power lines swaying due to wind (Barth et al., 1994), the ground clutter has a zero Doppler velocity shift. Bandpass filters can isolate clutter and weather signals as long as the weather signal has a non-zero velocity. As the weather signal approaches zero velocity, more sophisticated methodologies are needed to separate clutter from desired weather signals (Siggia and Passarelli, 2004).

For scanning weather radars, both the clutter and weather signals have Gaussian shape peaks that enables removing the clutter signal and recovering any overlapping weather signal. Within the Doppler velocity spectrum domain, the Gaussian model adaptive processing (GMAP) method (Siggia and Passarelli, 2004) uses the saved coherent and quadrature time-series observations (i.e., I and Q voltages) to calculate multiple spectra to adaptively determine the Gaussian shaped clutter and remove it from the Gaussian shaped weather signal. The GMAP methodology applied to time-domain calculations (called GMAP-TD) accounts for scanning radars utilizing staggered pulse repetition time (PRT) sequences (Nguyen and Chandrasekar, 2013). Since vertically pointing radars in mixed-phased clouds routinely observe signals from two hydrometeor types (e.g., liquid clouds and falling ice particles, Shupe et al., 2004; Kalesse et al., 2016), the GMAP method cannot be implemented to remove clutter without significantly modifying the GMAP logic. In addition, the time-series I and Q voltages needed to resample the spectra with different amplitude weightings are often not available for reanalysis from vertically pointing radars.

Receiving backscattered energy from moving trees and cars through antenna sidelobes is a common clutter problem with wind profilers (Barth et al., 1994). Due to the relatively large antenna beamwidths in wind profilers (e.g., 6-to-9 degrees for main beams and larger for sidelobes), clutter tends to be broad Gaussian-shaped features near zero velocity (May and Strauch, 1998). Birds and bats are often detected in higher frequency wind profilers (e.g. 915 MHz) with wavelet and time-domain Gabor



transforms being effective in identifying and removing these short-time duration targets (Jordan et al., 1997; Lehmann, 2012). In addition, the Gabor transform technique has been shown to improve wind profiler horizontal wind estimates (Bianco et al. 2013). In contrast to wind profilers, Ka-band cloud radars have very narrow beamwidths (on the order of 0.3°) such that Doppler velocity spectrum broadening due to horizontal motion through the radar beam is negligible (Shupe et al., 2008; Kollias et al., 2007). These

narrow beamwidths result in very narrow clutter peaks in the Ka-band cloud radar velocity spectra with insects appearing as very narrow spectral peaks (Luke et al., 2008).

    Recent studies have shown that velocity spectrum skewness provides information of drizzle onset (Kollias et al., 2011; Luke and Kollias, 2013; Acquistapace et al., 2017) and for deriving properties of ice clouds (Maahn et al., 2015; Maahn and Löhnert, 2017). Since there is a trade-off between temporal resolution and spectrum noise variance, the spectral moment estimates

tend to be noisy for short duration spectra (Giangrande et al. 2001; Luke and Kollias, 2013; Acquistapace et al. 2017). By shifting spectra to a reference velocity before averaging spectra, Luke and Kollias (2013) showed that spectrum skewness estimates improved and were more coherent in time and height.

    There is a long history of estimating multiple peaks in radar Doppler velocity spectra (Clothiaux et al., 1994). These multiple peaks need to be estimated before applying fuzzy logic (Cornman et al., 1998; Cohn et al., 2001; Morse et al., 2002),

neural network (Gardner and Dorling, 1998), or wavelet (Lehmann and Teschke, 2001) frameworks to discriminate atmospheric signals from clutter and radio interference. Estimating multiple peaks is a form of data reduction, or feature extraction, that can be used as inputs to algorithms that estimate boundary layer heights (Allabakash et al., 2017) or horizontal winds (Liu et al., 2017).

    This paper has the following structure. Section 2 describes the radar deployment and operating parameters of a US Department of Energy (DOE) Atmospheric Radiation Measurement (ARM) program Ka-band zenith pointing radar (KAZR)

installed at Oliktok Point, Alaska. Section 3 describes signatures of clutter and atmospheric signals observed in KAZR velocity spectra. Section 4 develops a clutter identification and mitigation method. This section also discusses how multi-path scattering from a nearby scanning radar antenna caused the clutter to have either approaching or receding radial motion. Section 5 describes a method to identify multiple peaks in the spectra and estimate high-order spectral moments. Section 5 also discusses a method of shifting individual spectra to the 15-s mean velocity before averaging. Section 6 provides concluding remarks. For completeness

and repeatability, Appendix A provides the equations to estimate high-order spectral moments.

## 2 Radar Observations

Since the early 1990s, the US Department of Energy (DOE) Atmospheric Radiation Measurement (ARM) program has deployed atmospheric observing systems around the globe to measure and characterize the radiative properties of the atmosphere (Mather and Voyles, 2013). The radiative properties of clouds are dependent on many factors including cloud composition, cloud thickness,

and temperature. Measurements from vertically pointing cloud radars, lidars, and radiometers provide the input observations needed to estimate and to better model the radiative properties of clouds (Clothiaux et al., 2000).

    In 2015, DOE installed their third ARM Mobile Facility (AMF-3) at Oliktok Point, on the North Slope of Alaska, which is approximately 264 km east-southeast of the long-term ARM North Slope of Alaska (NSA) core-observing site near Utqiaġvik (formally known as Barrow). The AMF-3 instrument suite includes a Ka-band (35 GHz) ARM zenith pointing radar (KAZR) and

a Ka/W-band (35/94 GHz) scanning ARM cloud radar (SACR, until September 2017) with both antennas installed on top of the same shipping sea-container as shown in Fig. 1. The SACR antennas are less than 2 m from the KAZR antenna with a direct line-of-sight to the KAZR feed-horn. As will be investigated in Section 3, the rotating SACR antennas are in the KAZR near-field



which caused the clutter from the stationary ground clutter targets to be Doppler shifted alternating between approaching and receding as the SACR antennas rotated above the KAZR antenna.

*(Place Figure 1 near here: Fig 1. Photo of KAZR and SACR)*

The KAZR and SACR at Oliktok Point became operational after an intensive Calibration, Grooming, and Alignment (CGA) field campaign conducted by the ARM Radar Engineering Group in October 2015. The KAZR operates in three modes: general mode (GE), medium mode (MD), and precipitation mode (PR). The initial operating parameters recorded 256-point Doppler velocity spectra and on 16-June-2016, the number of incoherent integrations were reduced in order to record 512-point spectra and maintain the same time-on-target (Table 1). Raw spectra used in this study are available on the DOE ARM Archive (ARM Climate Research Facility, 2015).

At Oliktok Point, oil refineries, pipelines, and powerlines within 2.5 km range are detected by KAZR as backscattered energy reflects back toward the radar and leaks into the radar system through antenna sidelobes. To mitigate the ground clutter observed in the KAZR spectra, a temporary clutter screen was installed around the KAZR antenna on 27-August-2016 (Fig. 1b). Thus, there are three different KAZR configurations partitioned by date: (prior to 16-June-2016) 256-point spectra with no clutter screen; (between 16-June-2016 and 26-August-2016) 512-point spectra with no clutter screen; and (after 26-August-2017) 512-point spectra with clutter screen.

## 3. Atmospheric and Non-Atmospheric Signal Signatures

Stationary ground clutter will appear in the Doppler velocity spectra near zero velocity. This section examines and quantifies the characteristics, or signatures, of KAZR ground clutter and KAZR atmospheric signals due to clouds and precipitating particles. Appendix A provides details of calculating spectral moments from raw velocity spectra.

### 3.1 Ground Clutter Contamination

**Figure 2** shows time-height cross-sections of measured radar reflectivity (Fig. 2a) and mean radial velocity (Fig.2b, positive values are approaching the radar) for 1 hour of observations starting at 12:00 UTC on 19-June-2016. For this figure, instead of imposing a user defined signal-to-noise ratio threshold to discriminate spectra with signal-plus-noise versus spectra with just noise, moments were estimated for only spectra containing at least three consecutive spectral points above the noise threshold. While the actual observations for this precipitation event extend above 6000 m, the vertical axis in Fig. 2 is limited to 2500 m to show details of the ground clutter signatures. There are four general areas of interest in this figure: one area contains atmospheric signals and the other three areas contain clutter signatures. The atmospheric signals are due to cloud and precipitating particles that are identifiable by reflectivities greater than approximately -10 dBZ and downward velocities greater than 1 m s$^{-1}$ reaching the surface after minute 25. There is also a brightband in reflectivity near 1500 m and a large gradient in downward motion near 1500 m indicating the melting of ice particles into raindrops (e.g., Williams et al., 1995).

*(Place Figure 2 near here: Fig 2. Time-height cross-sections of original KAZR Doppler velocity spectral moments.)*

Clutter is visible in Fig. 2a and 2b within two height ranges prior to minute 20. Clutter signatures are either below 600 m or within 1500-to-2000 m. In both height regions, the clutter reflectivity is nearly constant at each height and the radial velocity is near zero. Note that the radar continuously detects clutter within these two height ranges throughout the hour. The clutter signature is not visible after minute 20 because signal power from the cloud and precipitation is larger than the clutter power and the spectral peak picking routine is selecting the larger atmospheric peak. At a height near 500 m and minutes 30-to-60, there are intervals





when the clutter peak is larger than the atmospheric peak such that the spectral peak picking routine has selected the clutter peak instead of the atmospheric peak. These clutter peaks appear near minutes 31 and 50-to-57 and are distinguished by discontinuities in reflectivity and near zero radial velocities.

### 3.2 Drop in Power from Peak to Nearest Neighbour

There are significant differences in the characteristics of backscattered return power from distributed targets and from point targets (Mahafza, 2017). In the case of distributed hydrometeor targets, the hydrometeors have different sizes and velocities that are constantly moving within the radar pulse volume so that the radar-received backscattered power fluctuates from pulse-to-pulse (i.e., Swerling Type II targets). In addition, there is a distribution of different particle sizes falling at different velocities leading to a broad velocity distribution in the recorded Doppler velocity spectrum.

In contrast, received power return from stationary point targets is nearly constant from pulse-to-pulse with small random statistical fluctuations (i.e., Swerling Type 0 or V targets). The constant path-length between the radar and the target results in zero Doppler motion. In an ideal signal-processing environment, the stationary target in the time domain would transform into a delta-function of finite energy at zero velocity in the frequency domain. However, in real-world signal processors, the delta-function energy spreads over several velocity bins following a *sinc* function. The *sinc* function breadth and amplitude are determined by a

windowing function applied to the time-series before performing a fast-Fourier transform, and by the delta function amplitude (Mafazha 2017).

            In general, distributed hydrometeor targets produce broader velocity spectra than stationary targets. To explore these attributes in the recorded spectra, Fig. 2c shows the drop in received power from the velocity bin with peak magnitude to its directly neighbouring velocity bin expressed in units of dBm (i.e., power relative to 1 mW). Since there are two neighbouring velocity bins

bounding the peak value, all calculations use the largest power drop. Figure 2c shows that the power drop for the clutter signal is approximately 6 dBm (i.e., red colours) and occurs prior to minute 25, and near 500 m during minutes 31 and 51-57. For the spectra with clouds and precipitation, the drop in power is a distribution of values with a central value near approximately 2 dBm (i.e. blue colours). Figure 2c suggests that the power drop from the peak magnitude to the nearest neighbour is a good indicator of whether the spectrum peak is due to point-target scattering (approximately 6 dBm drop) or due to distributed hydrometeor target scattering

(less than 2 dBm drop).

            To explore details of how clutter signals appear in the recorded velocity spectra, **Figure 3a** shows a profile of spectra selected from Fig. 2 at 12:05:01 UTC on 19-June-2016. The radial velocity is on the abscissa and only extends from 1 m s$^{-1}$ upward (left side) to 1 m s$^{-1}$ downward (right side). The ordinate is height above the ground in meters and extends from the surface up to 800 m. The pseudo-colours represent received power in dBm. In nearly all range gates, ground clutter power is identifiable as an

increase in received power at zero velocity with additional power leaking into neighbouring velocity bins.

*(Place Figure 3 near here: Fig 3. Doppler velocity spectra profile with interpolation.)*

            The black line at 447 m in Fig. 3a indicates the height of the spectrum shown in Fig. 3c which contains both a clutter peak and an atmospheric peak due to cloud droplet particles (black line with pluses). A linear interpolation in linear units is performed across the three points centred about zero velocity and is shown in Fig. 3c with a red line and circles. It is important to note that

the 3-point interpolation only modifies power recorded at three velocity bins. Figure 3b shows stacked spectra after applying this 3-point interpolation to each spectrum. Note that this simple interpolation was sufficient to remove or suppress the clutter near zero velocity such that the atmospheric signal can be resolved.

            To illustrate the relative constant amplitude of the clutter signal with time, **Fig. 4a** shows 1774 consecutive spectra at 447 m height for hour 12 UTC on 19-June-2016 (which is the same hour shown in Fig. 2). Radial velocity is on the abscissas with



upward motion on the left and downward motion on the right. Time is on the ordinate with time increasing up the page. Pseudo-colours represent measured return power in dBm. A clutter peak near zero velocity is present during the whole hour while there are two fluctuating atmospheric signals at this height. A liquid cloud is present for most of the hour with updraft / downdraft magnitudes less than 0.5 m s$^{-1}$. After approximately minute 28, raindrops appear at this height with downward radial velocities

5   ranging from approximately 0.25 to over 4 m s$^{-1}$. The small magnitude power signals with upward motions after minute 28 are artefacts due to large magnitude downward power signals causing harmonics in the radar receiver. Figure 4b shows the spectra after applying the 3-point interpolation across the zero velocity. This simple 3-point interpolation is sufficient to remove the clutter peaks without disturbing the atmospheric signals.

*(Place Figure 4 near here: Fig 4. Time-series of radial velocity spectra.)*

## 4. Clutter Identification and Mitigation

As shown in Fig. 2 during the first 20 minutes of observations, without clutter peak mitigation, standard single-peak picking algorithms (e.g., Carter et al., 1995) will select ground clutter as a viable peak and will estimate the spectral moments of this clutter peak. If the clutter peak is in the middle of the atmospheric signal as in the example spectrum shown in Fig. 3c, then the estimated reflectivity will be biased high and the mean radial velocity will be biased toward zero velocity. If clutter mitigation is applied to

all spectra regardless of whether clutter signals are present, then low magnitude atmospheric signals centred on zero velocity could be eliminated from the data set. Thus, this section examines the power drop near zero velocity in order to establish a threshold to determine when and when not to apply clutter mitigation.

### 4.1 Clutter-to-Noise Ratio (CNR)

To determine whether the spectrum contains point-target signatures, three statistics are calculated for each spectrum: the drop in
power from the zero velocity bin to the nearest neighbour velocity bin ($P_{drop}$), the clutter-to-noise ratio (CNR), and the signal-to-noise ratio (SNR) of the decluttered spectrum. $P_{drop}$ is a clutter indicator while CNR quantifies the clutter power. **Figure 5a** shows a typical spectrum containing clutter power near zero velocity, collected at 00:14:33 UTC on 4-July-2016 at range 387 m. While the recorded spectrum extends to upward and downward Nyquist velocities of 5.9 m s$^{-1}$, this figure only shows radial velocities out to 0.4 m s$^{-1}$. A peak power of approximately -60 dBm occurs at zero velocity and $P_{drop}$ is approximately 6 dB. The thick

dashed line shows the 3-point interpolation with the light grey shaded area indicating the clutter power. The dark shaded area represents noise power. The CNR is defined as the clutter power (light grey shaded area but expressed in linear units) divided by total noise power (dark grey shaded area extended to Nyquist velocities and expressed in linear units) with CNR expressed in decibel units [dB].

*(Place Figure 5 near here: Fig 5. Selected spectra to illustrate clutter signal.)*

In Fig. 5a, notice the large power drops between the zero velocity bin and the first and second neighbouring velocity bins. The power drops are approximately 6 and 25 dBm, respectively. These large power drops are consistent with the expected *sinc* function from point targets. A large power drop to the second neighbouring bin is not always observed because either 1) the spectrum power falls below the noise threshold or 2) the power drop is masked by an atmospheric signal power as in the case shown in Fig. 5c.

After removing the clutter power using a 3-point interpolation (Fig. 5a), the spectral moments are estimated using the decluttered spectra (Fig. 5b) to determine whether or not the spectrum contains residual clutter or contains atmospheric signals. In Fig. 5b, the decluttered spectrum has eight consecutive spectral points above the noise threshold and is shaded medium grey. The





signal-to-noise ratio (SNR) of this residual peak is -8.3 dB, the largest magnitude power occurs at one of the velocity bins used in the 3-point interpolation (indicated with a filled circle), and this velocity bin is called the 'peak magnitude velocity'.

The spectrum shown in Fig. 5c contains both clutter and atmospheric signals (range 387 m at 17:14:31 UTC on 7-July-2016). The clutter peak is clearly identifiable in the spectrum. A thick dashed line shows the 3-point interpolation across zero velocity and the clutter power is shaded in light grey. Figure 5d shows the decluttered spectrum with the residual signal power and noise power indicated with the medium and dark grey shadings, respectively. For this spectrum, the residual peak SNR is 3.3 dB and the peak magnitude velocity (indicated with a filled circle) occurs away from either 3-point interpolation velocity bins and is associated with the atmospheric signal.

In order to determine when to apply the 3-point interpolation across zero velocity, we need to compare the drop in power in spectra with and without ground clutter. **Figure 6** shows a time-series of radial velocity spectra at a range of 987 m for the same 1-hour interval shown in Fig. 4a. There is no clutter in the raw spectra at 987 m shown in Fig. 6 so it can be used as a reference. Note that there are no saved spectra prior to minute 21 because the automated data reduction and archiving algorithm did not detect any spectral points (clutter or atmospheric signals) with power greater than the Hildebrand and Sekhon (1974) noise threshold.

*(Place Figure 6 near here: Fig 6. Time-series of radial velocity spectra at 987 m.)*

The power drop from zero velocity to the nearest neighbour $P_{drop}$ and CNR statistics were calculated for all spectra shown in Fig. 6. Only 210 spectra had a positive power drop with **Fig. 7a** showing a scatter plot of CNR vs. power drop (squares). The CNR vs. power drop for the simultaneous 210 spectra at 477 m (Fig. 4a) are shown in Fig. 7a using circles. The CNR for the spectra with and without ground clutter are approximately +10 dB and -25 dB, respectively. Figure 7b shows the power drop cumulative distribution functions (CDFs) and illustrates that clutter-free spectra have a broad distribution ranging from 0 to 4 dBm (dashed line) and the clutter spectra have a narrow distribution centred around 6 dBm (solid line). Note that all clutter spectra power drops are greater than 3 dBm (Fig. 7a) and that 90% of the clutter-free spectra have power drops greater than 3 dBm (Fig. 7b). While an exhaustive study could be undertaken to optimise a $P_{drop}$ threshold, this study found that a 2 dB power drop threshold identified nearly all clutter pixels at the expense of performing the 3-point interpolation across a limited number of spectra with atmospheric signals.

*(Place Figure 7 near here: Fig 7. Scatter plot of CNR vs. power drop and CDF power drop.)*

### 4.2 Ground Clutter Doppler Shift

Time-height clutter patterns occurred with a repeatable temporal cadence. Specifically, there were periods of narrow-symmetric clutter and periods of broader-asymmetric clutter. While the peak magnitude velocity rarely deviated from zero velocity, the asymmetry caused the mean velocity moment to deviate from zero velocity. **Figure 8** shows an hour's worth of observations on 3-July-2016 (hour 20 UTC) when no hydrometeors were above the radar. Figure 8a shows the clutter-to-noise ratio (CNR) [dB], Fig. 8b shows the residual peak SNR [dB] and Fig. 8c shows the residual peak magnitude velocity expressed in m s$^{-1}$. Note that the peak magnitude velocity alternates between negative and positive radial velocities suggesting that the stationary ground clutter has a Doppler motion component and is either receding or approaching the radar, respectively.

*(Place Figure 8 near here: Fig 8. Time-height cross-section of clutter statistics.)*

Figure 9 shows the pointing direction of the SACR scanning radar antennas (solid line) for minutes 20-to-30 during hour 20 UTC on 3-July-2016. During this 10-minute interval, the SACR antennas are rotating at 2°/minute in a clockwise direction. The white and grey shading in **Fig. 9** represents the column median residual peak magnitude velocity estimated from Fig. 8c and is either receding (white) or approaching (grey). There is a clear relationship between clutter Doppler motion and the SACR antenna pointing direction. As the SACR antennas complete one rotation, the clutter motion completes one receding-approaching cycle.





We postulate that the pulse-to-pulse change in path-length between the KAZR antenna and the stationary targets via multi-path reflections off the rotating SACR antenna caused the Doppler shift. The different durations and occurrences of residual peak magnitude velocities shown in Fig. 8c correspond to the different SACR scanning modes (not shown here). While a relationship between clutter occurrence and SACR scanning mode is interesting, the focus of this work is to identify and mitigate clutter

signatures occurring in the spectra.

*(Place Figure 9 near here: Fig 9. SACR antenna azimuth pointing direction.)*

**4.3 Clutter Mitigation Logic Diagram**

This section describes a clutter mitigation routine that identifies and removes static and non-static clutter signals from recorded velocity spectra. The development of this simple routine is after developing several complex routines with multiple thresholds and

multiple conditional statements. After systematically removing thresholds and conditionals, this final routine has only three conditionals:

1) Is power $P_{drop}$ greater than a threshold?
2) Are there enough spectral points above the noise threshold to estimate moments?
3) Is the residual peak magnitude velocity at either 3-point interpolation edge velocity?

**Figure 10** shows a flow diagram for the clutter mitigation routine. Starting with box #1, a single spectrum is loaded into the routine. The power drop from zero velocity to the nearest neighbour is calculated $P_{drop}$ (box #2). If $P_{drop}$ is greater than a threshold as derived from Figure 7b (box #3), then this spectrum is flagged to contain clutter and passed to box #5; else, this spectrum does not contain clutter (box #4) and is saved for future spectral and moment processing (box #13). If the spectrum gets into box #5, the 3-

point interpolation is performed across zero velocity and the residual moments are estimated in box #6.

*(Place Figure 10 near here: Fig 10. Clutter identification and mitigation flow diagram.)*

Box #7 requires that at least five consecutive spectral points have magnitudes greater than the noise threshold. This condition is an additional clutter filter that removes narrow clutter signals and also prevents high-order moments of velocity skewness and kurtosis to be calculated on narrow atmospheric signals. Without enough spectral points, this spectrum is flagged to

not contain any atmospheric signal (box #8) and is barred from being used in any temporal spectral averaging techniques discussed in Section 4 (box #9). If there are five or more spectral points above the noise threshold, then an evaluation is performed to see whether the residual peak magnitude velocity is at the edge of either 3-point interpolation edge velocity (box #10). If it is, then this residual likely still contains clutter (box #11) and the spectrum is barred from further analysis (box #9). If the peak magnitude velocity is different from either 3-point interpolation edge velocity, then this spectrum likely contains an atmospheric signal (box

#12) and is saved for future spectral and moment estimations (box #13).

To illustrate the performance of the clutter identification and mitigation routine, the same spectra used to construct Fig. 2 were processed through the flow diagram shown in Fig. 10. **Figure 11** shows the decluttered spectra calculations of reflectivity (Fig. 11a), mean radial velocity (Fig. 11b), and power drop from peak power to nearest neighbour (Fig. 11c). Figure 11 shows a vertically thin cloud layer just below 500 m that was not visible in Fig. 2 because of the contaminating ground clutter. This is

consistent with the spectra analysis shown in Figs. 3 and 4 that showed an oscillatory cloud layer near 500 m. In Fig. 11, a 3x3 time-height continuity filter was applied such that centre pixels were removed when domains did not have at least three neighbouring pixels (See Appendix B).

*(Place Figure 11 near here: Fig 11. Time-height cross-section of decluttered moments.)*





## 5. Multiple Peaks and High-Order Spectral Moments

After identifying and removing clutter in the effected spectra, this section describes how to identify multiple-peaks in the spectra, how to estimate high-order moments for each spectral peak, and how to construct 15-s average spectra using a 'shift-then-average' procedure.

### 5.1 Identifying Multiple Peaks

One advantage of processing radar velocity spectra is that different hydrometeor habits can be identified by their velocity signatures. For example, **Fig. 12a** shows a velocity spectra profile when both cloud particles and ice particles are occurring in the same height between 500 to 800 m. This profile was collected on 15-October-2016 at 11:55:55 UTC with the pseudo-colours representing received power in dBm. By eye, we can see two return power patterns in the spectra profile. One pattern is limited in

height between 500 to 800 m and has downward motions between 0 and 0.4 m s$^{-1}$ that correspond to signals from cloud particles. Another return signal extends from the top of the panel to the surface with downward motions ranging from 0 to 2 m s$^{-1}$. While more analysis is needed to determine whether these return signals are from liquid or ice phase particles, we can confidentially state that faster falling particles are larger than the cloud particles that are confined to the 500 to 800 m range.

*(Place Figure 12 near here: Fig 12. Spectra profile with multiple peaks.)*

Super imposed on the spectra in Fig. 12a are the mean velocity $V_{mean}$ (black vertical ticks) and +/- one velocity spectrum standard deviation $V_{sig}$ (horizontal black lines) estimated assuming that one spectral peak exists in each spectrum (see Appendix A for spectrum moment equations). These moments are known as 'single peak' moments and have a long lineage in vertically pointing radar research (see Carter et al., 1995) and the DOE ARM community (see Clothiaux et al., 2000). We can see that the single-peak moments do not represent the dual-peak nature of the recorded spectra. To overcome this limitation, multiple spectral

peaks are identified with $V_{mean}$ and $V_{sig}$ estimated for each peak and shown in Fig. 12d. The black symbols are from single peak moments while the blue and red symbols are from sub-peaks and separate peaks identified in the spectra.

Identifying multiple peaks (Luke and Kollias, 2013) is a process of identifying boundaries, or integration limits, which will be used in the spectrum moment equations. To help describe how boundaries are identified, **Fig. 13** shows how single peaks, sub-peaks, and separate peaks are identified in example spectra pulled from heights 807 and 777 m in Fig. 12. Table 2 provides a

description of the three types of peaks. Every spectrum with at least five consecutive spectral points above the noise threshold will have a single peak. However, not every spectrum with a single peak will have sub-peaks or separate peaks.

*(Place Figure 13 near here: Fig 13. Velocity spectra with multiple peak integration limits.)*

The spectrum from 807 m (Fig. 13a) has two spectral peaks. The peak on the right is the most significant peak because it contains the spectral point with the largest magnitude. The integration limits for the single peak extend over all consecutive points

above the noise threshold. The triangles in Fig. 13a indicate the single peak integration limits and $V_{mean}$ and $V_{sig}$ are plotted near 0.8 m s$^{-1}$ downward velocity. Since there are spectral points below the noise threshold between the single peak and the left peak, the left peak is called a separate peak. Circles indicate the integration limits for this separate peak. The separate peak must have five consecutive spectral points above the noise threshold.

Figure 13b shows the spectrum from 777 m. The single peak is very broad and extends from approximately 0.25 m s$^{-1}$

upward to 1.3 m s$^{-1}$ downward as indicated with the triangles. Sub-peaks are peaks within the single peak separated by a local minimum, or valley, that has at least 6 dB of concavity. The circles indicate the integration limits for two sub-peaks in Fig. 13b. The $V_{mean}$ for all three peaks are shown with triangles and circles with $\pm V_{sig}$ shown with lines. Similar to the other peaks, sub-peaks must have five consecutive spectral points above the noise threshold.





### 5.2 High-Order Spectral Moments

After identifying integration limits for all spectral peaks, the high-order moments are calculated for each peak using the equations shown in Appendix A. The spectral moments range from the signal-to-noise ratio (the zeroth moment) to the velocity spectrum kurtosis (the fourth moment).

The scatter-plot profiles on the right side of Fig. 12 show the spectral moments of reflectivity and velocity skewness for the single, sub-, and separate peaks. The top row shows only the single peak moments while the bottom row shows moments from different peaks. Note that if sub-peaks exist in a spectrum, then the single peak moments are not plotted in the bottom row. The reflectivity vertical structure for the multiple peaks (Fig. 12e) shows both a continuous pattern with height and two patterns that are limited in height. The continuous pattern mimics the single peak reflectivity pattern shown in Fig. 12b and has a local maximum

near 400 m. The two height-limited patterns occur near 700 and 1400 m where there are two distinct hydrometeor populations in the spectra profile (Fig. 12d). Near 700 m, the smaller reflectivity values correspond to the cloud particles with mean velocities near 0.2 m s$^{-1}$ downward.

       With regard to the velocity skewness, the single peak estimates (Fig. 12c) show large negative values below 800 m with maximum value near 600 m. A negative velocity skewness indicates that the long distribution tail is on the negative velocity side of the peak, which is upward motion in this dataset. Yet, Fig. 12f shows near-zero velocity skewness for the sub-peaks between

500 and 800 m. This suggests that large magnitude single peak velocity skewness could indicate the existence of multiple sub-peaks. Yet, after identifying sub-peaks, velocity skewness represents the asymmetry of each individual spectral peak. Thus, single peak velocity skewness could be used to identify the existence of multiple sub-peaks and moments from multiple peaks should be used to perform quantitative microphysical analyses.

### 5.3 Shift-then-Average Spectra

As discussed in Luke and Kollias (2013), the velocity spectrum skewness can be a noisy estimator due to velocity bin-to-bin spectrum power fluctuations. To improve the velocity spectrum skewness estimate, Luke and Kollias (2013) suggested shifting consecutive spectra to a common reference, averaging the shifted spectra, and then estimating the velocity spectrum variance and skewness. Shifting the spectra before averaging reduces the spectrum broadening and smearing due to vertical air motion variability

that occurs during the averaging interval (Giangrande et al., 2001).

       The method of shifting all spectra to line-up all peak magnitude velocities appeared to work well for the maritime drizzle clouds (Luke and Kollias 2013), but it did not work well with Arctic mixed-phase clouds observed at Oliktok Point because the peak magnitude sometimes jumped to a different spectral peak during the 15-s integration interval. To overcome this occasional issue, the spectra were shifted to the 15-s mean velocity. Specifically, shift-then-average processing consisted of nine steps

performed at each range gate:

1) Estimate the single peak mean velocity for each 2-s spectrum $V_{mean}^{2s}$
2) Incoherently average (no shifting) all spectra within a 15-s interval
3) Identify the single peak in this 15-s averaged spectrum
4) Estimate the single peak mean velocity $V_{mean}^{15s,inc}$ and velocity spectrum variance $V_{var}^{15s,inc}$ for this 15-s incoherent

averaged spectrum
5) Shift each 2-s spectrum by $V_{mean}^{shift}$ so that: $V_{mean}^{2s} + V_{mean}^{shift} = V_{mean}^{15s,inc}$
6) Average the shifted spectra
7) Identify multiple peaks in shifted-then-averaged spectra
8) Estimate high-order moments for each identified peak

9) Save all multiple peak moments as well as the incoherent averaged spectra mean velocity $V_{mean}^{15s,inc}$ and velocity
            spectrum variance $V_{var}^{15s,inc}$



As an example of the shifting process, Fig. 14a shows eight spectra (thin lines) collected at 447 m on 15 October 2016 during the 15-s interval starting at 11:55:45 UTC. The average of these eight spectra is shown with a thick line. The spectral moments of this averaged spectrum are listed in Table 3. The mean radial velocity $V_{mean}^{15s,inc}$ is used as the reference velocity to shift

each spectrum as shown in Fig. 14b (thin lines). The mean of these shifted spectra is shown in Fig. 14b with a thick line and the spectral moments are listed in Table 3. The shift-then-averaged spectrum (Fig. 14b) has a narrower breadth than the simple incoherent averaged spectrum (Fig. 14a) which is confirmed with the values listed in Table 3. In addition, velocity spectrum skewness and kurtosis become more pronounced and have larger magnitudes after shifting and then averaging the spectrum. One benefit of shifting the individual spectra to the 15-s mean velocity before averaging is that there is an additional spectrum breadth

estimate available for turbulence studies. Namely, the spectrum breadth of the shift-then-average spectrum does not have the broadening caused by 2-s velocity shifts during the 15-s interval.

## 6. Concluding Remarks

This study is a combined science and engineering effort designed to improve high-order moments estimated from Ka-band (35-GHz) vertically pointing radar Doppler velocity spectra by developing three different signal-processing methods. First, a

15 decluttering method identifies and removes clutter in the Doppler spectra. Since hard targets produce narrow spectral peaks, identifying clutter is based on large power drops from the zero-velocity bin to the nearest velocity bin neighbour. A linear interpolation is performed on spectra identified as contaminated with clutter. All spectra void of clutter and those mitigated of clutter are used in the subsequent processing methods. As an interesting side note, we found that a rotating antenna within 2 meters of the Ka-band vertically pointing radar is causing the clutter to be Doppler shifted. We postulate reflected waves bouncing off the

20 rotating antenna cause the path length between the Ka-band antenna feed horn and the stationary targets to change from pulse-to-pulse, which artificially changes the target range during the 2-sec dwell producing a Doppler shift.

The second method developed in this study identifies multiple peaks and calculated high-order moments for each single peak, sub-peak, and separate peak. Identifying multiple peaks is a process of identifying the integration limits that are used in the high-order moment calculations. The high-order moments included velocity spectrum skewness and kurtosis. This work found that

spectrum skewness from the single spectral peak is a good indicator of whether two hydrometeor populations are present in the radar pulse volume. Yet, the sub-peak and separate peaks are symmetric with skewness estimates near zero. This suggests a two-step process of using single peak velocity skewness as an indicator of possible multiple peaks and multiple-peak moments for quantitative studies.

The third method developed in this study is shifting individual 2-s spectra during 15-s intervals to the mean velocity

before averaging the spectra. This shift-then-average method improves the velocity spectrum skewness estimates by removing the spectrum turbulent broadening effects at the 2-s temporal scale.

*Data availability*. Original raw KAZR spectra are available on the DOE ARM archive, doi: 10.5439/1025218 (ARM Climate Research Facility, 2015). Also, six months (May-Oct 2016) of Oliktok Point KAZR spectra were processed using the clutter

mitigation, multiple peak, and shift-then-average techniques discussed in this study and are available at the DOE ARM archive as Evaluation Data.



### Appendix A – High-Order Spectral Moment Equations

This appendix defines the equations used to calculate high-order spectral moments. The spectral moments are calculated for each recorded radial velocity spectrum $S(i)$ which has a length $N_{pts}$, indices $i$ range from 1 to $N_{pts}$, and has units of Watts. The radial velocity $v(i)$ for this spectrum has velocity resolution $\Delta v$, velocity range from receding Nyquist velocity ($v(1) = -V_{Nyquist}$) to

maximum approaching velocity ($v(N_{pts}) = V_{Nyquist} - \Delta v$), and has units of m s⁻¹. Zero velocity has the index $i_{zero\ velocity} = \frac{N_{pts}}{2} + 1$. Note that the sign of the radial velocity is negative for receding targets ($sgn(v) < 0$) and positive for approaching targets ($sgn(v) > 0$). This physical notation insures that falling particles have positive radial velocities that correspond to positive physical diameters.

Noise statistics are determined after sorting the spectrum magnitudes using the method described in Hildebrand and

Sekhon (1974). Essentially, this method sorts all spectrum values and then determines a threshold that divides the data into either noise-only data or noise-plus-signal data. Using the noise-only data, three noise statistics are defined: mean noise $n_{mean}$, noise standard deviation $n_{std}$, and noise threshold $n_{threshold}$ which is the largest magnitude noise-only data point.

Before estimating the spectral moments, integration limits, or summation limits for discretely sampled spectra, need to be determined. Following Carter et al. (1995), the largest magnitude spectral value is determined ($S(i_{max}) = \max(S)$) and a logical

pointer is positioned at this velocity $v(i_{max})$. The left integration index $i_{left}$ is determined by moving the pointer down the left side of the spectrum until the spectrum magnitude is less than the noise threshold. Since the integration limit needs to start above the noise threshold, the left index is incremented so that $S(i_{left}) > n_{threshold}$. The right integration limit is determined in a similar way by starting at the largest magnitude value and progressing the pointer down the right side of the spectrum. Thus, spectral moments are calculated using the consecutive spectra and velocities from $S(i_{left})$ and $v(i_{left})$ to $S(i_{right})$ and $v(i_{right})$.

As discussed in the section 5, identifying multiple spectral peaks is a procedure to identify integration limits for each single peak, sub-peak, and separate peak. After identifying the left and right indices, $i_{left}$ and $i_{right}$, for each peak, the following moments are calculated for each peak.

Noise Power:

$$n_{power} = n_{mean} N_{pts} \qquad \text{(A1)}$$

Signal to noise ratio:

$$(SNR)_{dB} = 10 log \left[ \frac{\sum_{i_{left}}^{i_{right}} (S(i) - n_{mean}) \Delta v}{n_{mean} N_{pts} \Delta v} \right] \quad \text{[dB]} \qquad \text{(A2)}$$

Reflectivity weighted mean velocity:

$$V_{mean} = \frac{\sum_{i_{left}}^{i_{right}} S(i) v(i) \Delta v}{\sum_{i_{left}}^{i_{right}} S(i) \Delta v} \quad \text{[m s}^{-1}\text{]} \qquad \text{(A3)}$$

Velocity spectrum variance:

$$V_{var} = \frac{\sum_{i_{left}}^{i_{right}} S(i)(v(i) - V_{mean})^2 \Delta v}{\sum_{i_{left}}^{i_{right}} S(i) \Delta v} \quad \text{[m}^2\text{ s}^{-2}\text{]} \qquad \text{(A4)}$$

Velocity spectrum standard deviation:

$$V_{sig} = [V_{var}]^{0.5} = \left[ \frac{\sum_{i_{left}}^{i_{right}} S(i)(v(i) - V_{mean})^2 \Delta v}{\sum_{i_{left}}^{i_{right}} S(i) \Delta v} \right]^{0.5} \quad \text{[m s}^{-1}\text{]} \qquad \text{(A5)}$$



Velocity spectrum skewness:

$$V_{skewness} = \frac{1}{V_{sig}^3}\left[\frac{\sum_{i_{left}}^{i_{right}} S(i)(v(i)-V_{mean})^3 \Delta v}{\sum_{i_{left}}^{i_{right}} S(i)\Delta v}\right] \quad \text{[dimensionless]} \tag{A6}$$

Velocity spectrum kurtosis:

$$V_{kurtosis} = \frac{1}{V_{sig}^4}\left[\frac{\sum_{i_{left}}^{i_{right}} S(i)(v(i)-V_{mean})^4 \Delta v}{\sum_{i_{left}}^{i_{right}} S(i)\Delta v}\right] \quad \text{[dimensionless]} \tag{A7}$$

5   Spectrum peak magnitude $S_{peak}$ and index $i_{peak}$:

$$S_{peak} = S(i_{peak}) = max(S) \tag{A8}$$

Velocity at spectrum peak magnitude

$$V_{peak} = V(i_{peak}) \tag{A9}$$

10   **Appendix B – 3x3 Time-Height Continuity Filter**

A 3x3 time-height continuity filter can be applied to processed moments to remove observations that were not continuous in time and height. For every 3x3 (time-by-height) matrix of observations, each pixel was assigned to be either a valid or invalid observation. If the centre pixel is valid and there were not enough valid neighbouring observations in the 3x3 matrix, then the centre pixel was set to invalid. In this study, at least 3 neighbouring pixels were needed to retain a valid centre pixel.



*Author contribution*. CRW: clutter filter and multiple peak code development; MM, JH, and GB: testing and evaluation products.

*Competing interests*. The authors declare that they have no conflict of interest.

*Disclaimer*. The authors declare no disclaimers.

*Acknowledgements*. This research received funding through the DOE Atmospheric System Research (ASR) program under
5  awards DE-SC0013306 and DE-SC0014294. We recognize and appreciate the work of field technicians, especially radar
technician Todd Houchens of Sandia National Laboratory, deployed year-round to Oliktok Point tasked with keeping these
instruments running in extremely harsh and challenging weather conditions. This research was supported by the Office of
Biological and Environmental Research of the U.S. Department of as part of the Atmospheric Radiation Measurement (ARM)
Climate Research Facility, an Office of Science scientific user facility.




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

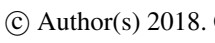



**Tables**

*Table 1*. **Operating parameters for AMF-3 KAZR deployed at Oliktok Point, Alaska, from 1-October-2015 through 31-October-2017 (at the time of publishing, the radar was still operating at Oliktok Point, Alaska). Operating modes included General Purpose (GE), Medium Sensitivity (MD), and Precipitation (PR) modes. Tabulated parameters include: pulse repetition frequency ($PRF$) [Hz], inter-pulse period ($IPP$) [μsec], number of points in FFT ($N_{FFT}$), number of averaged spectra (also known as number of incoherent integrations) ($N_{ave}$), unambiguous velocity ($V_{unambiguous}$) [m s⁻¹], velocity resolution (Δv) [m s⁻¹], range to first range gate [m], range resolution [m], time-on target (which is calculated using $IPP\ N_{FFT}\ N_{ave}$) [s], and time between samples [s]. Radar operating modes changed on 16-June-2016 at 14 UTC. A clutter screen was installed on the antenna on 27-August-2016.**

| Parameter | before 16-June-2016 (on/after 16-June-2016) | | |
| --- | --- | --- | --- |
| | GE | MD | PR |
| Pulse Repetition Frequency ($PRF$) [Hz] | 2777 | 2777 | 2777 |
| Inter-Pulse Period ($IPP$) [μsec] | 360 | 360 | 360 |
| Number of points in FFT ($N_{FFT}$) | 256 (512) | 256 (512) | 256 (512) |
| Number of incoherent integrations $N_{incoh}$ | 18 (9) | 18 (9) | 4 (2) |
| $V_{unambiguous}$ [m s⁻¹] | 5.97 | 5.97 | 5.97 |
| $\Delta v$ [cm s⁻¹] | 4.67 (2.33) | 4.67 (2.33) | 4.67 (2.33) |
| Range to first range gate $R_1$ [m] | 40 (57) | 707 (737) | 40 (57) |
| Range resolution $\Delta R$ [m] | 30 | 30 | 30 |
| Time-on Target $t_{target} = IPP\ N_{FFT}\ N_{incoh}$ [s] | 1.66 | 1.66 | 0.37 |
| Time between samples $t_{sample}$ [s] | 2.0 | 2.0 | 2.0 |



*Table 2.* **Attributes and integration limits for three spectral peak regimes: single peak, sub-peak, and separate peak. All spectral peaks need at least 5 consecutive spectral points with magnitudes greater than the noise threshold (Hildebrand and Sekhon 1974).**

| Peak Name | Attributes | Integration limits |
|---|---|---|
| Single Peak | Contains the largest magnitude spectral point<br><br>(Every valid spectrum has a single peak) | Determined by noise threshold |
| Sub-Peak | Sub-peaks are within integration limits of the single peak | Determined by noise threshold or<br>Determined by valley of at least 6 dB between sub-peaks |
| Separate Peak | There are spectrum points below the noise threshold separating this peak from the single peak | Determined by noise threshold |





*Table 3.* **Spectral moments of averaged spectrum after averaging eight (8) spectra using two different methods. All eight (8) spectra were collected during 15-s interval on 15-Oct-2016 between 11:55:45 and 11:56:00 UTC and are shown in Fig. 14. The averaged spectrum was constructed by averaging individual spectra as shown in Fig. 14a. The mean radial velocity from this averaged spectrum is used as the reference velocity. The shifted-then-averaged spectrum was constructed by first shifting individual spectra to a reference mean radial velocity and then averaging as shown in Fig. 14b.**

| Spectral Moment | Averaged Spectrum | Shifted-then-Averaged Spectrum |
|---|---|---|
| Signal-to-Noise Ratio, $(SNR)_{dB}$ [dB] | 23.24 | 23.24 |
| [1]Mean Radial Velocity, $V_{mean} = V_{mean}^{15s,inc}$ [m s$^{-1}$] | 1.22 | 1.22 |
| Velocity spectrum standard deviation, $V_{sig}$ [m s$^{-1}$] | 0.32 | 0.28 |
| Velocity spectrum skewness, $V_{skewness}$ [unitless] | -1.11 | -1.44 |
| Velocity spectrum kurtosis, $V_{kurtosis}$ [unitless] | 4.63 | 5.38 |

[1]15-s averaged spectrum mean velocity used as reference velocity in shifted-then-averaged procedure.



**Figures**

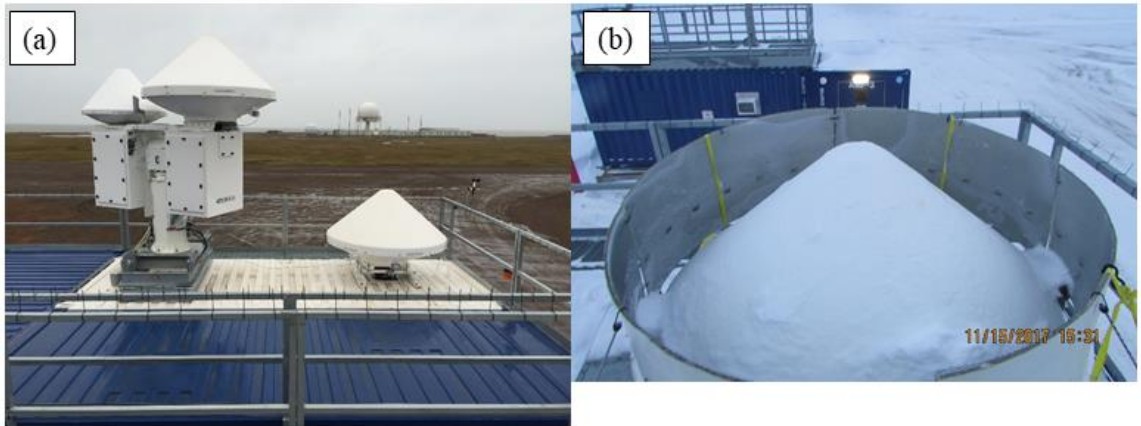

**Figure 1: (a) Photo of AMF-3 Ka-/W-band Scanning ARM Cloud Radar (Ka/W-SACR) antennas (left) and Ka-band ARM Zenith Radar**
5  **(KAZR) antenna (right) as deployed at Oliktok Point, Alaska (Photo credit: Gijs de Boer). (b) Photo of clutter screen mounted around KAZR antenna with snow on the KAZR radome (Photo credit: Joe Hardin)**





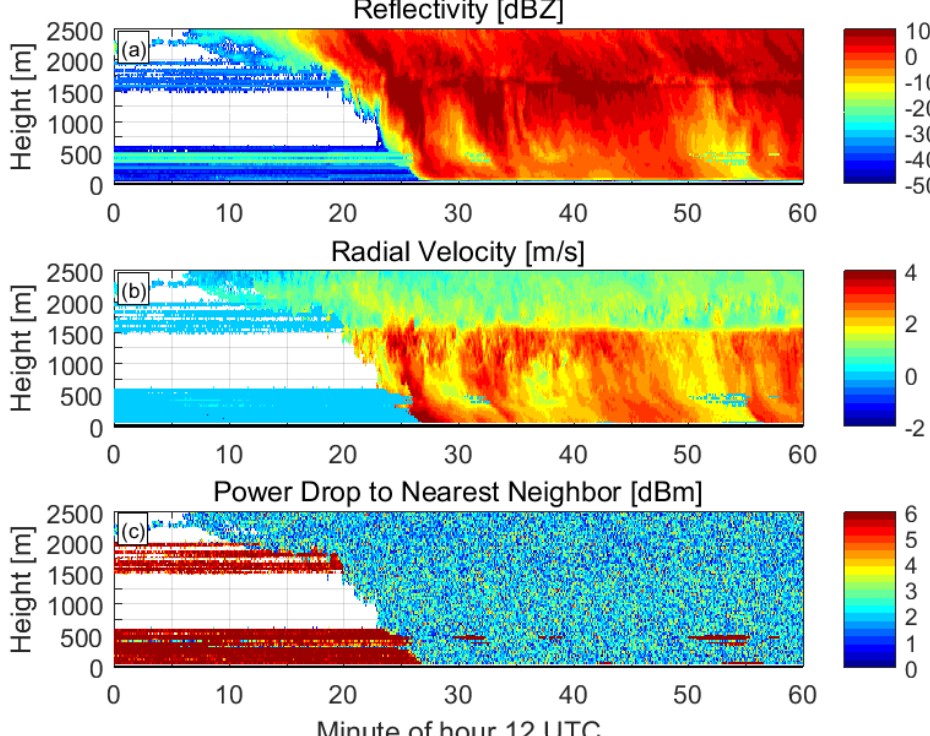

**Figure 2: Time-height cross-sections of original KAZR Doppler velocity spectral moments and attributes from 19-June-2016 during hour 12 UTC. (a) Reflectivity [dBZ], (b) mean radial velocity of dominant single peak [m s⁻¹] (positive values are approaching the radar), and (c) maximum power drop from peak magnitude to either nearest neighbour [dB]. At least three consecutive spectral points needed to be above the noise threshold before estimating the moments. Large power drop from peak to nearest neighbour is an indicator that clutter peak was selected as dominant peak.**




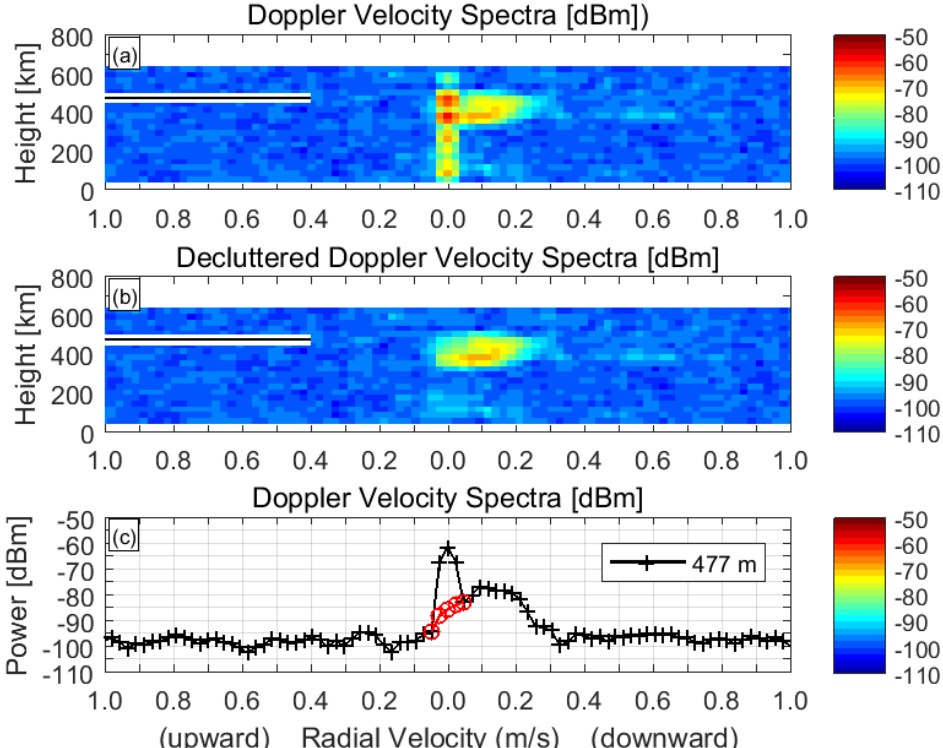

**Figure 3: Doppler velocity spectra for profile collected on 19-June-2016 at 12:05:01 UTC. (a) Original Doppler velocity spectra at each range gate as a function of radial velocity. (b) Similar to panel (a) expected spectra were interpolated across DC (zero velocity) to mitigate clutter signal. (c) Original spectrum (black line and pluses) and decluttered spectral points (red line and circles) at 447 m range (black line in (a) and (b)).**





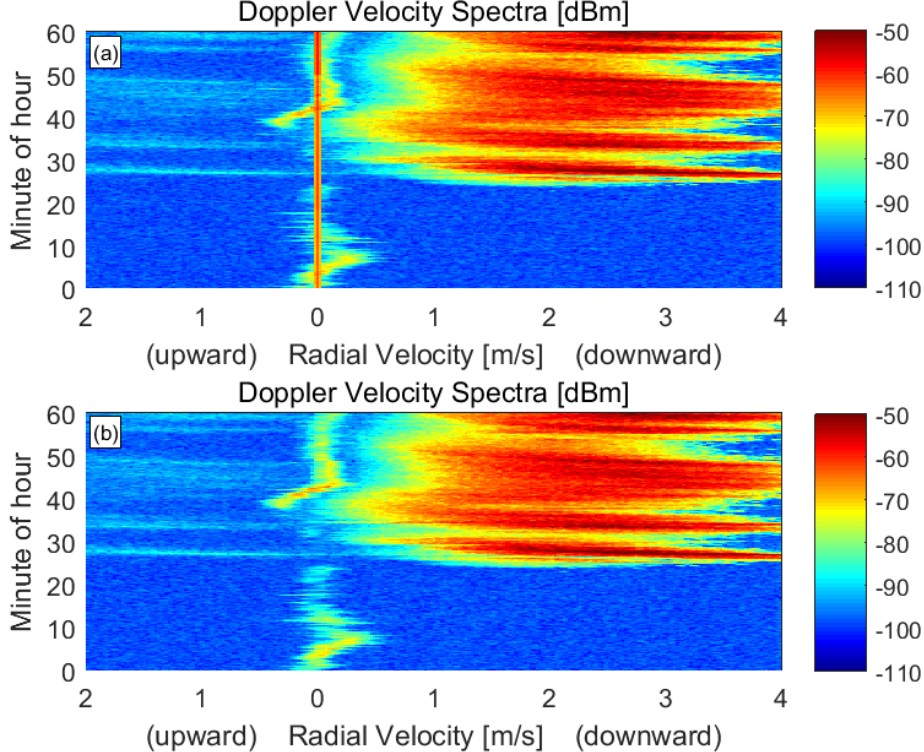

**Figure 4: Time-series of radial velocity spectra on 19-June-2016 during hour 12 UTC at 447 m range. First spectrum of hour is at the bottom of the panel (minute 0) and last spectrum of the hour is at the top of the panel (minute 60). Horizontal axis is upward (left side) and downward (right side) radial velocity. (a) Original spectra and (b) decluttered spectra.**





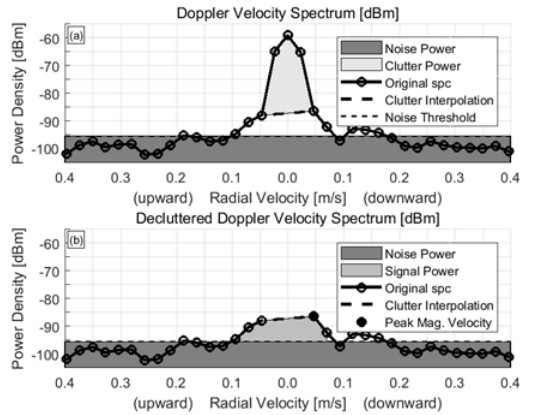
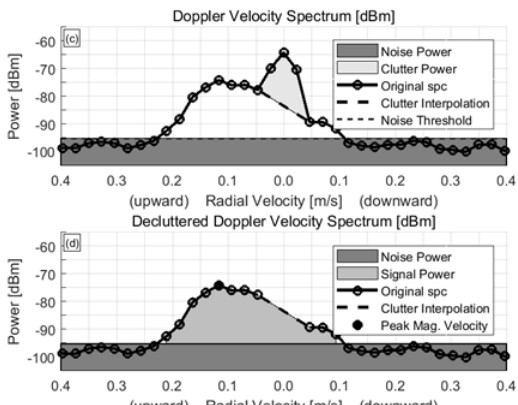

**Figure 5: Selected spectra to illustrate clutter signal, 3-point interpolation, and residual signal power. (a) Original spectrum from 4-July-2016 at 00:14:33 UTC at 387 m range with clutter power shaded in light grey. (b) Same spectrum as in (a) except 3-point linear interpolation across zero velocity to remove clutter power with residual signal peak power shaded in middle grey colour. The noise power is shaded the darkest grey in all panels. Panels (c) and (d) are similar as panels (a) and (b) except spectra were collected on 7-July-2016 at 17:14:31 UTC. The solid circle in panels (b) and (d) indicate the velocity of the peak magnitude in the residual spectra. Due to this peak magnitude velocity, the residual signal in panels (b) and (d) are deemed residual clutter and atmospheric signal, respectively.**





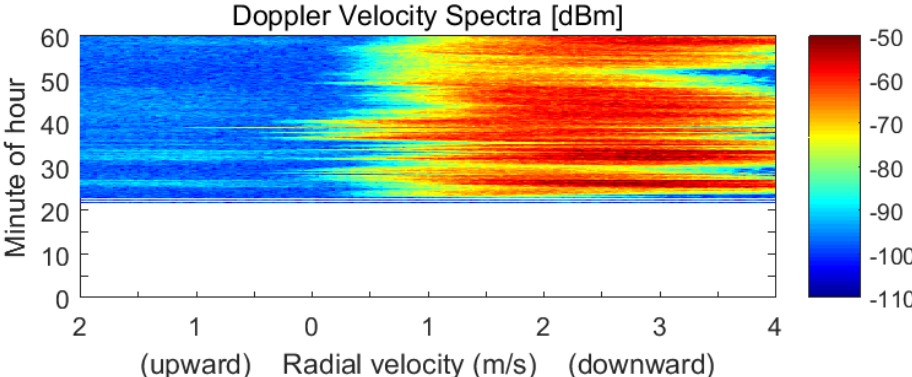

**Figure 6: Similar to Fig. 4a except for 987 m range. In contrast to range gate at 447 m shown in Fig. 4, this range gate does not contain ground clutter for this hour and does not contain any atmospheric signal before minute 21. Since no signal was detected above the noise threshold before minute 21, the data reduction and storage algorithm did not save any spectra before minute 21.**



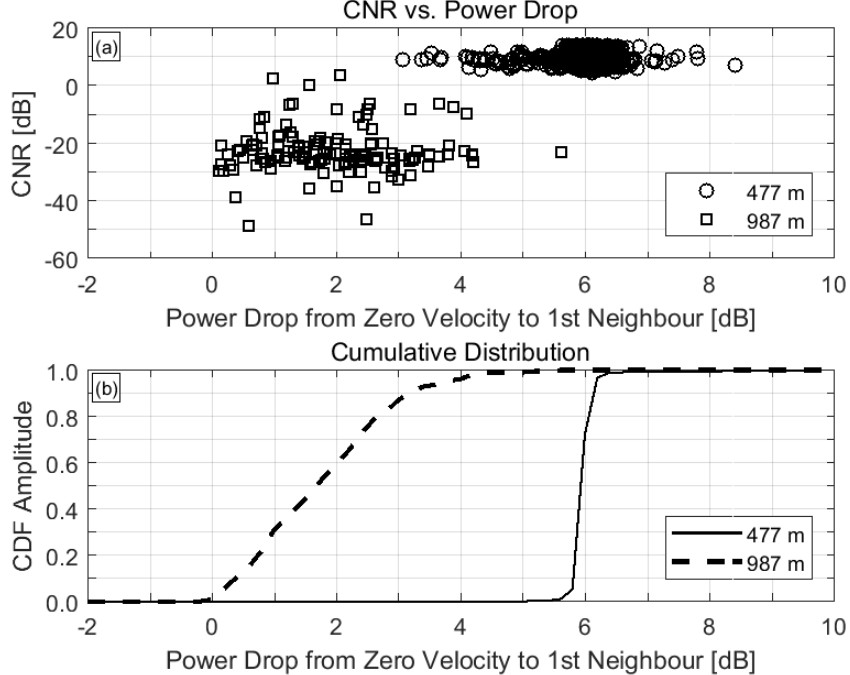

Figure 7: (a) Scatter plot of clutter-to-noise ratio (CNR) vs power drop $P_{drop}$ from zero velocity to nearest neighbor for the same profiles at ranges 477 m (circles) and 987 m (squares) for hour 12 of 19-June-2016. Spectra at 477 m contained clutter and spectra at 987 m did not have clutter. (b) Cumulative distribution function (CDF) of power drop for estimates shown in (a). These estimates were derived from spectra shown in Fig. 4a and Fig. 6. There are 210 simultaneous samples during this rain event.



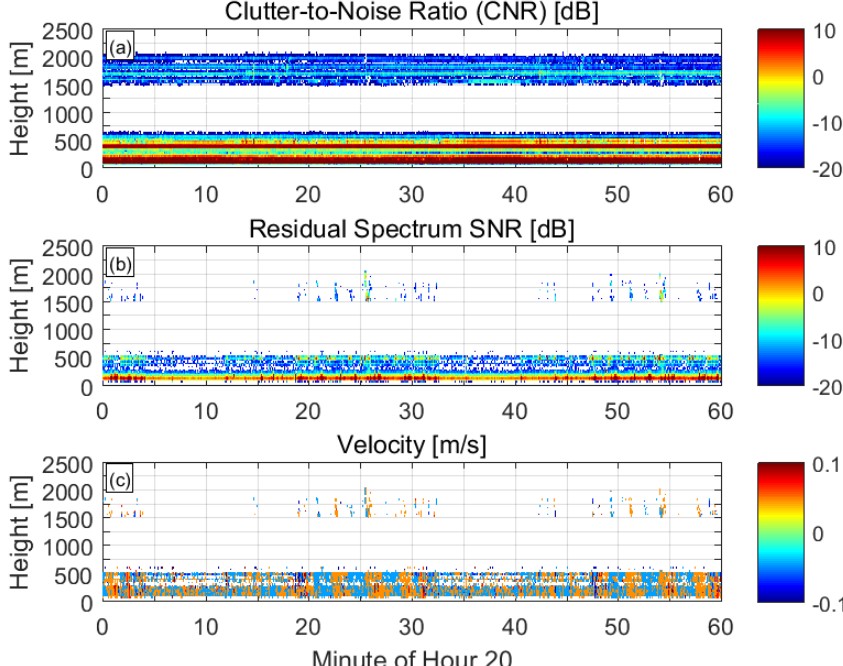

**Figure 8: Time-height cross-section of clutter statistics from spectra collected on 3-July-2016 during hour 20 UTC. (a) Clutter-to-noise**
5    **ratio (CNR) based on clutter power from three points centred around zero velocity (see Fig. 6), (b) signal-to-noise ratio of residual peak**
**after removing clutter peak, and (c) velocity of spectral peak in the residual spectrum indicates a skewness of the residual peak. Positive**
**peak velocities correspond to targets approaching the radar.**



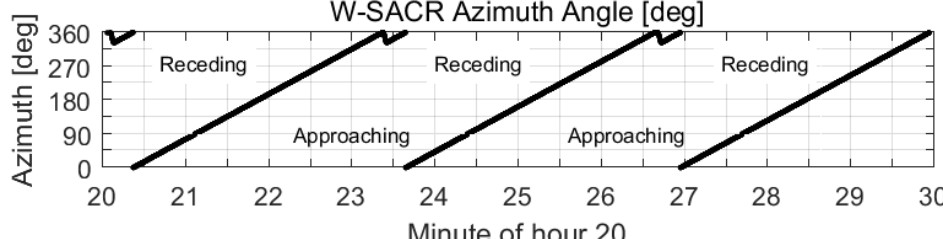

**Figure 9: Diagram showing SACR antenna azimuth pointing direction (solid line) and Doppler shift of residual clutter (shading) for 10 minutes of observations shown in Fig. 7. With a rotation rate of 2 °/min, the Ka/W-SACR antenna completed a rotation every 3 minutes.**
5     **The residual clutter Doppler shift contained two-phases per rotation indicating the antennas receding and approaching**



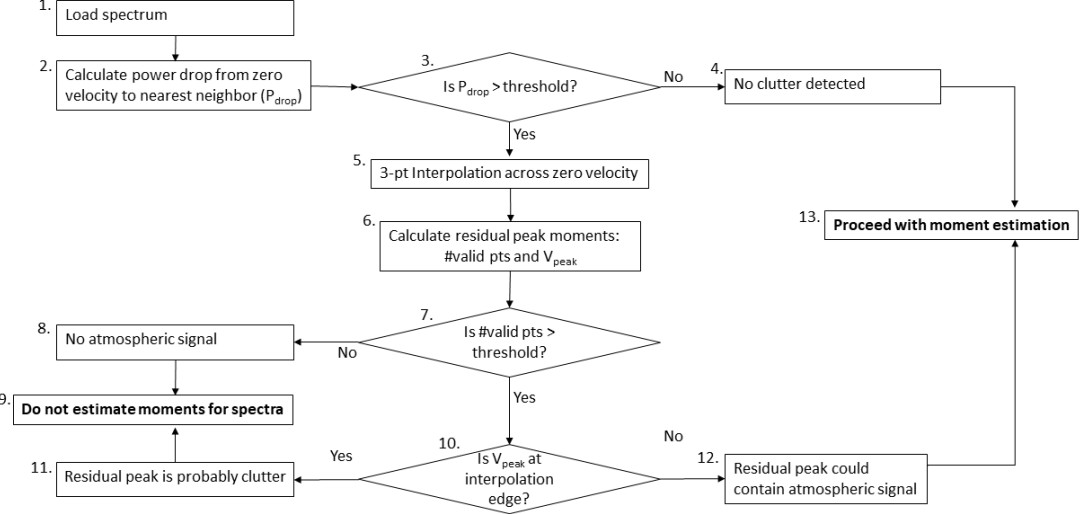

**Figure 10: Clutter identification and mitigation flow diagram. Processing is performed on individual spectra without knowledge of clutter being identified in neighbouring spectra.**



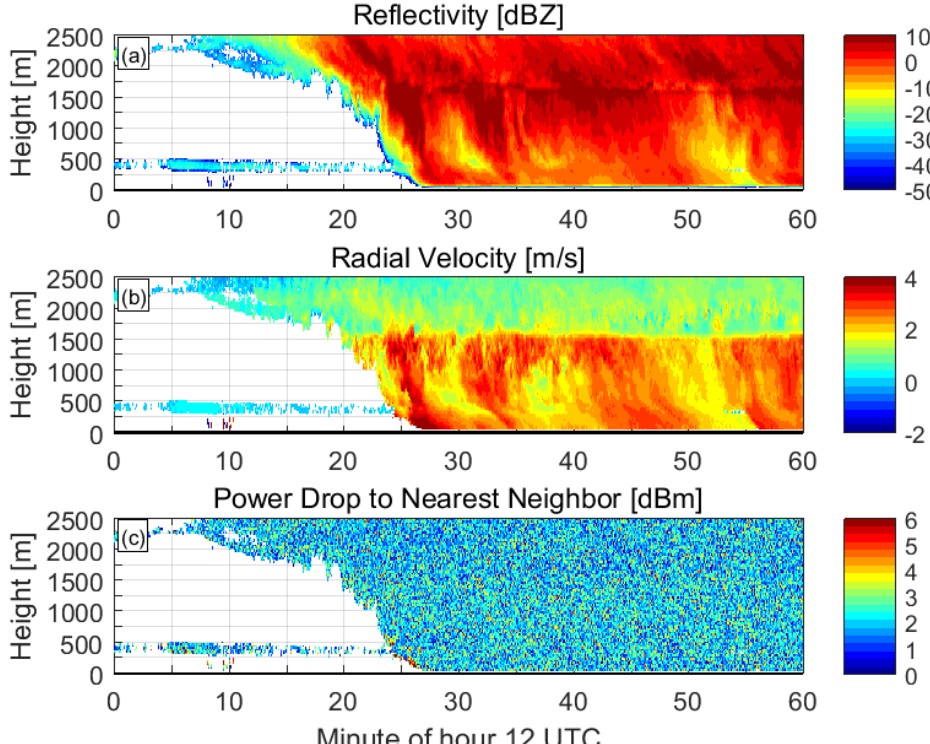

**Figure 11: Similar to Fig. 2 except spectra were decluttered (see section 3) before estimating (a) reflectivity, (b) mean radial velocity, and (c) power drop from peak power to nearest neighbour. After estimating moments in each profile, a 3x3 filter was applied to remove outliers.**





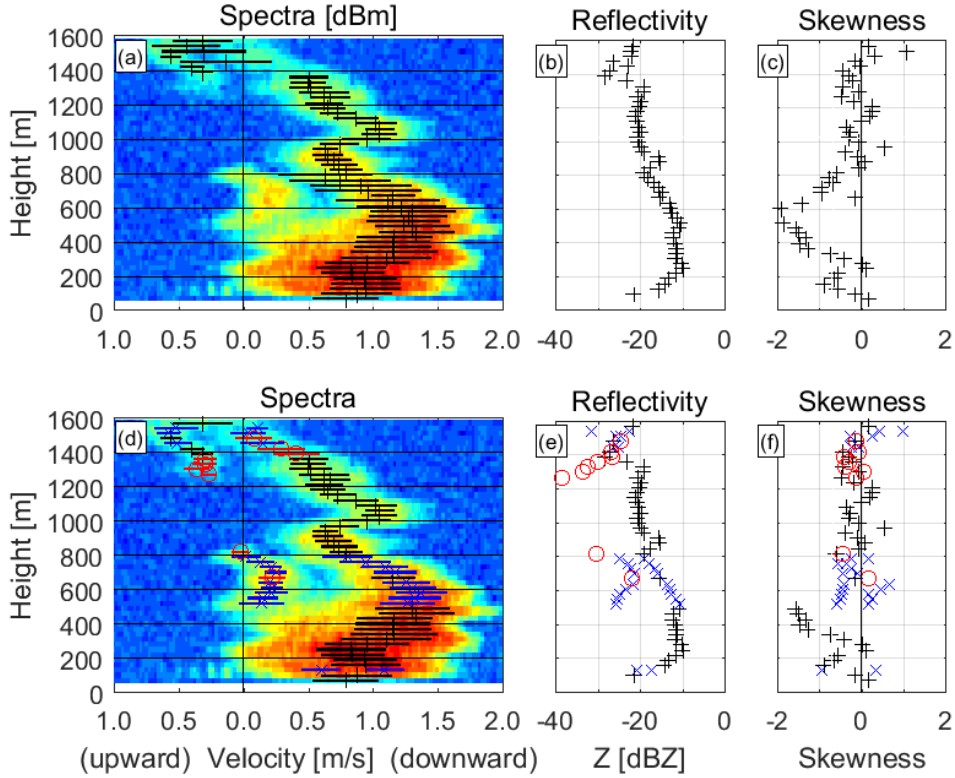

**Figure 12: Profile of spectra and moments collected on 15-Oct-2016 at 11:55:55 UTC. Top row corresponds to single-peak moments and bottom row corresponds to multiple-peak moments. Top row: (a) Pseudo-colour represent spectral power [dBm] with mean velocity shown with black pluses and +/- spectrum bread shown with black lines, (b) reflectivity, and (c) velocity skewness. Bottom row: similar as top row, except blue symbols represent sub-peaks and red symbols represent separate peaks.**





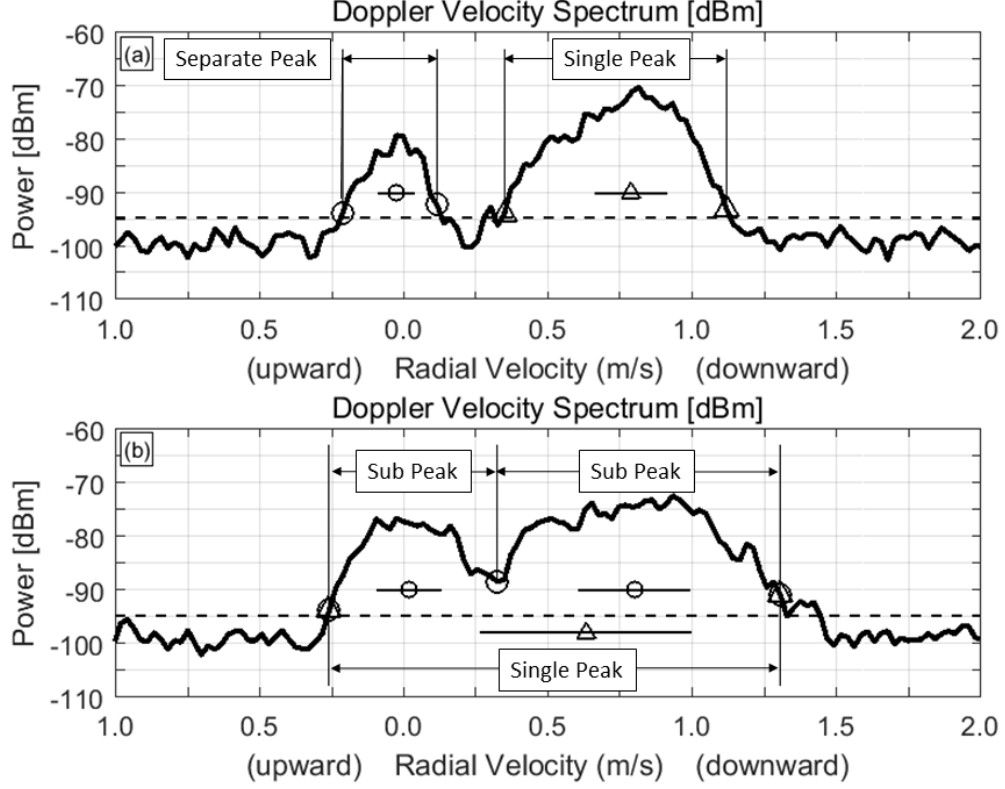

**Figure 13: Radial velocity spectra on 15-October-2016 at 11:55:55 UTC at ranges (a) 807 m and (b) 777 m. The spectrum in (a) contains two peaks separated with spectral power below the noise threshold. The single peak is the dominant peak due to the larger peak**
5  **amplitude. The spectrum in (b) contains a single peak that spans from approximately 0.25 m s⁻¹ upward to 1.3 m s⁻¹ downward. This single peak contains two sub-peaks with a valley (or local minimum) near 0.3 m s⁻¹ downward.**





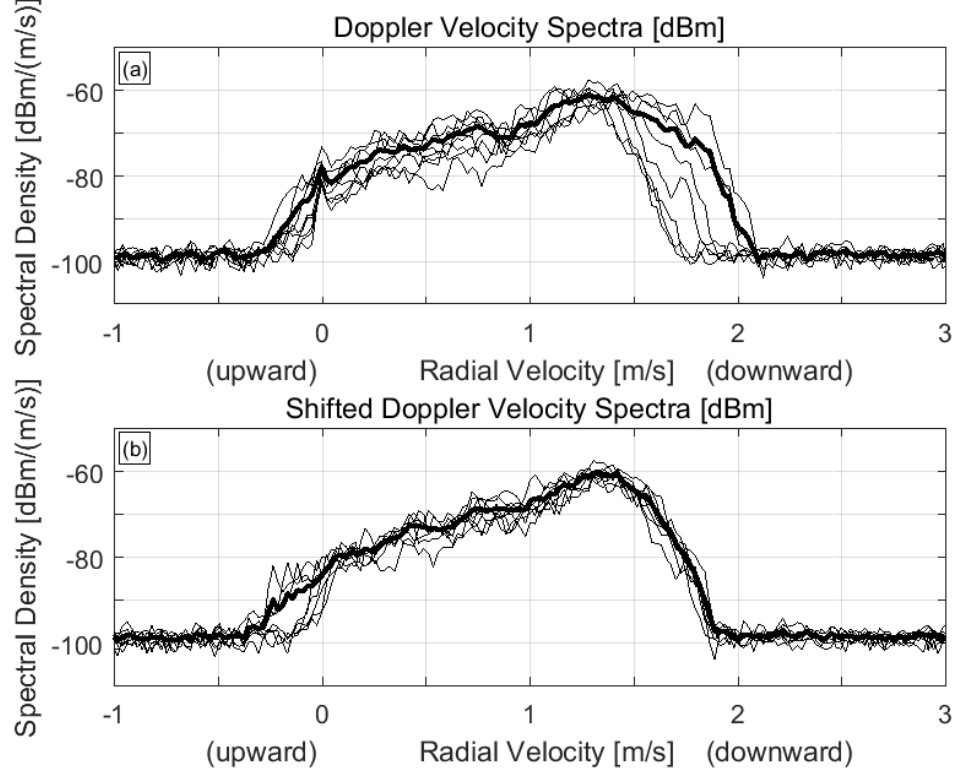

**Figure 14: Eight radial velocity spectra collected on 15 October 2016 during 15-s interval starting at 11:55:45 UTC at 447 m range. (a) The eight spectra (thin lines) were averaged to form an averaged 15-s spectrum (thick line). The spectral moments for the averaged spectrum are calculated with the mean radial velocity used as the reference velocity. In (b), each spectrum is shifted to have the same reference velocity (thin lines) and the mean value (thick line) is the shifted-then-averaged 15-s spectrum. The spectral moments are tabulated in Table 3.**