# Peer review of "Clutter Mitigation, Multiple Peaks, and High-Order Spectral Moments in 35-GHz Vertically Pointing Radar Velocity Spectra"

_Atmospheric Measurement Techniques, 2018_

## Referee Comment (RC1) · Anonymous Referee #2 · 2 May 2018

**Full review of Williams et al., AMT 2018**

**General comments**

The study titled "Clutter Mitigation, Multiple Peaks, and High-Order Spectral Moments in 35-GHz Vertically Pointing Radar Velocity Spectra" by Christopher R. Williams et al. describes three techniques used to improve the quality of high-order radar moments determined from 35GHz vertically pointing cloud radar Doppler spectra. This technical paper focuses on the detailed description and benefit of the processing methods. The first method can be regarded as pre-processing step as it shows how to remove ground clutter returns in the radar Doppler spectra caused by antenna side lobes and reflections of a nearby scanning cloud radar. In the second technique, the identification of multiple spectral peaks and the determination of their radar moments are presented. The third method focuses on improving the quality of higher-order moments (spectrum width, skewness, and kurtosis). It was demonstrated that 2-s temporal scale turbulent broadening can be removed by shifting all 2-s spectra to a common 15-s mean Doppler velocity before averaging.

**Open question:**

I would suggest the manuscript to be published after minor revision. The authors should address the following points:

**Major comments**

Abstract: You mention cloud and drizzle particles causing non-zero skewness. Since you do not explicitly mention what kind of clouds you are analyzing, the reader might think you focus on liquid-only clouds. This is misleading as you actually look at mixed-phase clouds, too. Please mention in the abstract to which clouds your technique can be applied to. Also specify briefly, if the described three techniques will be/are implemented in routine data processing of the ARM program or if new data products are planned or older ones enhanced by your methods. In technical papers, the challenge is always to provide detailed description of the methodologies while at the same time maintaining readability. This is often reached by splitting long sentences into two. Please check where this can be done.

Some of the thresholds described in the flow diagram (Fig10) seem a bit random:
p.8 line 22: Shouldn't the requirement of "number of spectral points above noise threshold" be a function to the spectral resolution thus be different for let's say the 256 vs 512 FFT points?
p.8 line 36: Why did you opt for "at least three neighboring pixels" in the 3x3 time-height continuity filter? Did you try other thresholds and if so, did results differ much?

p.9 line 36: How did you come up with a 6dB "valley" between the most significant peak and the subpeak? Did you try other thresholds and if so, did results differ much? Is this threshold based on radar forward simulations or empirically-based? Shupe et al., 2004 for example described that their peak-picking criteria were empirically based. They state that "For two continuous modes above the noise to be considered distinct modes, the saddle point between the peaks must be lower than 65% of the lowest of the two peaks from the noise level." – Did you try this instead of a fixed 6dB threshold?

In general, it would be desirable if you motivate the choice of your thresholds, compare your threshold values against literature values and discuss differences/advantages of your thresholds.

p.10 line 29: Again, please motivate why you chose 15-s integration intervals.

Please specify the thresholds for moment estimation: mean or max. noise floor?

**Minor comments**

p.1 Line 13: The phrasing is a little bit misleading: It sounds as if only the first method is applied to KAZR data.

p.1 Line 18: Unclear if "unique peak" refers to noise-floor separated peak. Please rephrase to clarify.

p.1 Line 19 etc.: Explain why you use the term "breadth" instead of spectrum "width"

p.1 Line 20f: Last sentence of the paragraph explains the third method (in the previous sentence) again. – For the abstract, I suggest leaving out this last sentence or merging the two sentences to save space.

p.1 Line 27: I suggest rephrasing to "…indicator of possible multiple hydrometeor populations"

p.2 Line 24ff: I suggest replacing "weather signal" with "hydrometeor signal"

p.4 Line 25: switch "only" and "for"

p.7 Line 22-24: This sentence is unclear. – Please rephrase.

p.8 Line 9-10: This sentence is unclear. – Please rephrase.

p.8 line 14: How can the residual peak magnitude velocity be at "either" 3-point interpolation edge velocity?: The middle velocity of the 3-point interpolation is not an edge velocity. – Please rephrase for clarity.

p.9 line 12: Did you have a look at the radar spectral LDR signature to discriminate spherical (likely liquid) particles from non-spherical ice particles?

---

## Referee Comment (RC2) · Anonymous Referee #1 · 15 May 2018

Review – Clutter Mitigation, Multiple Peaks, and High-Order Spectral Moments in 35-GHz Vertically Pointing Radar Velocity Spectra

This study presents techniques to process 35 GHz cloud radar vertical spectra, such that high order moments may be estimated, and thus microphysical and dynamical information retrieved. The paper first demonstrates the effectiveness of a linear interpolation method to remove clutter with zero Doppler shift, using a nearest neighbour power drop characteristic signature for point and distributed targets. The authors then discuss the identification of multiple spectral peaks, and finally a "shift-then-average" technique to improve the quality of high order moment estimation.

[Figure]

I suggest the manuscript is published after consideration of the following points.

General comments

Page 3 paragraph 2 – this paragraph discusses potential clutter sources for wind profilers which are not an issue at Ka band. It should be noted here that wind profilers operate at a range of frequencies, and thus at a range of sensitivities to clutter. Birds and bats are not a problem at lower VHF frequencies for example.

Page 3 line 15 – insects are mentioned here as presenting with narrow spectral peaks, but not mentioned again in the paper. Can the techniques described be applied to insect interference?

Page 6 paragraph 4 (near line 20) – how did you decide only 3 points should be interpolated across? Is this a function of the number of spectral points?

Page 8 line 20 – I think it should read power drops LESS than 3 dBm for clutter-free spectra. I am also curious as to why you discuss 3 dBm as the demarcation, and then choose 2 dBm for the threshold? The discussion indicates this it to be conservative, but perhaps this point could be made more clearly?

In general, the use of thresholds should be justified.

Technical corrections

Page 3 line 3 – "vertically point" should read "pointing"

Page 3 line 14 – first reference to KAZR, and thus should be defined as it is on page 4 lines 9 & 10

Page 4 line 13 – sentence does not make sense, perhaps remove "the clutter from the stationary ground clutter targets"

Page 4 line 23 – remove "from nearby structures"

Page 6 lines 13 & 25 – "abscissas" should be abscissa

Page 6 line 19 – remove either "in linear units" or "linear interpolation"

Page 6 line 24 – should read "illustrate the relatively"

Page 8 line 5 – "away from either 3-point interpolation" should read "away from THE 3-point interpolation"

Page 13 line 14 – "calculated high-order moments" should be calculates
* * *

---

## Author Comment (AC1) · 2 Aug 2018

**Author Replies to Reviewer #1 Comments**

Manuscript Title: Clutter Mitigation, Multiple Peaks, and High-Order Spectral Moments in 35-GHz Vertically Pointing Radar Velocity Spectra

Authors: Christopher R. Williams, Maximilian Maahn, Joseph C. Hardin, Gijs de Boer

We thank both reviewers for taking the time from their busy schedules to read and provide valuable feedback to our manuscript. All comments were thoughtful and lead to changes to the manuscript. Thank you.

Please find below specific replies to reviewer #1 comments. The packet of documents uploaded to AMT includes this document as well as the revised manuscript with and without Track Changes.

{*Reviewer comments are italic*. Author comments are indented and in regular font and.}

Reviewer #1.

Thank you for your helpful and insightful comments.

**General Comments**

*Page 3 paragraph 2 – this paragraph discusses potential clutter sources for wind profilers which are not an issue at Ka band. It should be noted here that wind profilers operate at a range of frequencies, and thus at a range of sensitivities to clutter. Birds and bats are not a problem at lower VHF frequencies for example.*

> This is a good point to highlight to the reader. A couple sentences were added to this paragraph. (See page 3, line 3 in the Track Change version.)

*Page 3 line 15 – insects are mentioned here as presenting with narrow spectral peaks, but not mentioned again in the paper. Can the techniques described be applied to insect interference?*

> Yes, the techniques can be applied to identify and remove insect scattering. The text in the first paragraph of the Conclusion was modified to mention that insects can be removed using these techniques. (See page 12, lines 3 to 14 in the Track Change version.)

*Page 6 paragraph 4 (near line 20) – how did you decide only 3 points should be interpolated across? Is this a function of the number of spectral points?*

Good point. Text was added to clarify that the 3-pt interpolation is based on the characteristics of this dataset and the reader should adapt this value to their dataset. (See page 6, line 36 in the Track Change version.)

*Page 8 line 20 – I think it should read power drops LESS than 3 dBm for clutter-free spectra. I am also curious as to why you discuss 3 dBm as the demarcation, and then choose 2 dBm for the threshold? The discussion indicates this it to be conservative, but perhaps this point could be made more clearly?*

*In general, the use of thresholds should be justified.*

Yes, the logic should be 'less' and not 'greater'. Text has been corrected. Also, the threshold was changed to 3 dBm to match the text and should be adjusted to match different radar datasets. Regarding thresholds in general, text has been added to mention that thresholds are dependent on the radar dataset, and that the code used in this study is available for download as supplemental material. Figure 11 was re-generated after using a threshold 3 dBm. (See page 7, line 26 and Fig. 11 in the Track Change version.)

**Technical Corrections**

*Page 3 line 3 – "vertically point" should read "pointing".*

Corrected.

*Page 3 line 14 – first reference to KAZR, and thus should be defined as it is on page 4 lines 9 & 10*

Corrected.

*Page 4 line 13 – sentence does not make sense, perhaps remove "the clutter from the stationary ground clutter targets"*

Sentence modified. (See page 4, line 4 in the Track Change version.)

*Page 4 line 23 – remove "from nearby structures"*

Text removed.

*Page 6 lines 13 & 25 – "abscissas" should be abscissa*

Corrected (See page 6, line 2 in the Track Change version.)

*Page 6 line 19 – remove either "in linear units" or "linear interpolation"*

Corrected (See page 5, line 35 in the Track Change version.)

*Page 6 line 24 – should read "illustrate the relatively"*

Corrected ()

*Page 8 line 5 – "away from either 3-point interpolation" should read "away from THE 3-point interpolation"*

Corrected ()

*Page 13 line 14 – "calculated high-order moments" should be calculates*

Corrected ()

---

## Author Comment (AC2) · 2 Aug 2018

**Author Replies to Reviewer #2 Comments**

Manuscript Title: Clutter Mitigation, Multiple Peaks, and High-Order Spectral Moments in 35-GHz Vertically Pointing Radar Velocity Spectra

Authors: Christopher R. Williams, Maximilian Maahn, Joseph C. Hardin, Gijs de Boer

We thank both reviewers for taking the time from their busy schedules to read and provide valuable feedback to our manuscript. All comments were thoughtful and lead to changes to the manuscript. Thank you.

Please find below specific replies to reviewer #2 comments. The packet of documents uploaded to AMT includes this document as well as the revised manuscript with and without Track Changes.

{*Reviewer comments are italic*. Author comments are indented and in regular font and.}

Reviewer #2.

Thank you for your helpful and insightful comments.

**Major Comments**

*Abstract: You mention cloud and drizzle particles causing non-zero skewness. Since you do not explicitly mention what kind of clouds you are analyzing, the reader might think you focus on liquid-only clouds. This is misleading as you actually look at mixed-phase clouds, too. Please mention in the abstract to which clouds your technique can be applied to. Also specify briefly, if the described three techniques will be/are implemented in routine data processing of the ARM program or if new data products are planned or older ones enhanced by your methods. In technical papers, the challenge is always to provide detailed description of the methodologies while at the same time maintaining readability. This is often reached by splitting long sentences into two. Please check where this can be done.*

  a. Liquid-only vs. mixed-phased clouds: Good suggestion. Text was added to the manuscript stating the multiple peaks can be applied to both types of cloud systems. (See page 1, line 24 in Track Change version.)
  b. Implementing code as routine data processing in ARM program. Text was added to the manuscript describing how this code is benefiting ARM data products. (See page 1, line 30 in Track Change version.)
  c. Splitting long sentences into two sentences. Thank you. Several sentences were broken into shorter sentences.

*Some of the thresholds described in the flow diagram (Fig10) seem a bit random: p.8 line 22: Shouldn't the requirement of "number of spectral points above noise threshold" be a function to the spectral resolution thus be different for let's say the 256 vs 512 FFT points?*

Yes, the thresholds do seem a bit random because they were selected for the Oliktok Point data set after many tests that are not presented in the manuscript. Text was added to the manuscript to describe the trade-offs a reader will have to make when they process their own datasets. (See page 8, line 34 in Track Change version.)

*p.8 line 36: Why did you opt for "at least three neighboring pixels" in the 3x3 time-height continuity filter? Did you try other thresholds and if so, did results differ much?*

Three neighboring pixels was chosen because it was found as the lowest value that efficiently removed remaining clear sky clutter. We found that the 3x3 time-height continuity filter helps remove isolated pixels that occur when clouds or precipitation are not nearby.

This comment made us look at the logic of applying the 3x3 time-height filter to the whole dataset. Since the 3x3 time-height continuity filter is not part of the actual clutter routine described in Fig. 10, we decided to make the 3x3 time-height filter a QC flag added to the netCDF output data file. The user can choose to apply this additional QC flag to mask data before their analysis, or, they can develop their own time-height continuity filter from the decluttered moments. Text was modified to reflect this change in the body and for fig. 11 caption. (See page 9, line 17 and page 13, line 3 in the Track Change version.)

*p.9 line 36: How did you come up with a 6dB "valley" between the most significant peak and the subpeak? Did you try other thresholds and if so, did results differ much? Is this threshold based on radar forward simulations or empirically-based? Shupe et al., 2004 for example described that their peak-picking criteria were empirically based. They state that "For two continuous modes above the noise to be considered distinct modes, the saddle point between the peaks must be lower than 65% of the lowest of the two peaks from the noise level." – Did you try this instead of a fixed 6dB threshold?*

Our experience with working with multiple peak spectra is that the valley between peaks is highly dependent on the bin-to-bin power fluctuations across the spectrum. These fluctuations are due to both noise fluctuations and signal power fluctuations. The bin-to-bin fluctuations of the Oklitok Point data tended to have a mean of about 2 dB with fluctuations exceeding 4 dB. These fluctuations are explored for the drop from the peak magnitude in Fig. 7a, but were not addressed for all points between the integration limits. Text was added to highlight how a user needs to evaluate their

dataset to determine the valley threshold. (See page 10, line 22 in the Track Change version.)

It is interesting to examine the Shupe et al. 2004 valley threshold. The 65% of the lower peak corresponds to a valley threshold of 10 log(0.65) = -1.9 dB. Before looking for the valley, Shupe et al. applied a 3-point boxcar averaging window to the spectrum to remove bin-to-bin spectrum variability. Thus, it is hard to compare our un-smoothed spectrum 6-dB threshold to Shupe et al.'s smoothed spectrum 1.9 dB valley threshold.

*In general, it would be desirable if you motivate the choice of your thresholds, compare your threshold values against literature values and discuss differences/advantages of your thresholds.*

Good suggestion. This suggestion was the primary reason to provide the MATLAB code used to process the Oliktok Point KAZR spectra as supplemental material. If users had the same code used in this study, they could start with these thresholds and modify them for their dataset.

Text was added to the manuscript to motivate threshold choices and describing the availability of the MATLAB code (See page 1, line 33; page 3, line 28; page 7, line 29; page 8, line 26; page 9, line 1; page 9, line 16; page 12, line 35 in the Track Change version.)

*p.10 line 29: Again, please motivate why you chose 15-s integration intervals.*

Text was added to the manuscript describing the compromise between 4-, 15-, and 60-s integration intervals. (See page 11, line 13 in the Track Change version.)

*Please specify the thresholds for moment estimation: mean or max. noise floor?*

The text was added to the manuscript clarifying the use of mean noise and maximum noise. (See page 13, line 12 in the Track Change version.)

**Minor Comments**

*p.1 Line 13: The phrasing is a little bit misleading: It sounds as if only the first method is applied to KAZR data.*

Modified Text. (See page 1, lines 12, 13, and 15 in the Track Change version.)

*p.1 Line 18: Unclear if "unique peak" refers to noise---floor separated peak. Please rephrase to clarify.*

Modified Text. (See page 1, lines 18 and 19 in the Track Change version.)

*p.1 Line 19 etc.: Explain why you use the term "breadth" instead of spectrum "width"*

Changed to "variance". We purposely do not use spectrum width because of the extra factor of 2 in the spectrum width definition, spectrum width = 2*sqrt(variance), where variance is the velocity spectrum variance. (See page 1, line 20 in the Track Change version.)

*p.1 Line 20: Last sentence of the paragraph explains the third method (in the previous sentence) again. – For the abstract, I suggest leaving out this last sentence or merging the two sentences to save space.*

Good suggestion. Sentences merged together. (See page 1, line 21 in the Track Change version.)

*p.1 Line 27: I suggest rephrasing to "…indicator of possible multiple hydrometeor populations"*

Done. (See page 1, line 29 in the Track Change version.)

*p.2 Line 24: I suggest replacing "weather signal" with "hydrometeor signal"*

Done. (See page 2, line 24 in the Track Change version.)

*p.4 Line 25: switch "only" and "for"*

Done. (See page 4, line 27 in the Track Change version.)

*p.7 Line 22 - 24: This sentence is unclear. – Please rephrase.*

Rewrote the sentence. (See page 7, line 27 in the Track Change version.)

*p.8 Line 9 - 10: This sentence is unclear. – Please rephrase.*

Rewrote the sentence. (See page 8, line 17 in the Track Change version.)

*p.8 line 14: How can the residual peak magnitude velocity be at "either" 3---point interpolation edge velocity?: The middle velocity of the 3---point interpolation is not an edge velocity. – Please rephrase for clarity.*

Thank you for your comment. Rewrote the sentence. (See page 8, line 23 in the Track Change version.)

p.9 line 12: Did you have a look at the radar spectral LDR signature to discriminate spherical (likely liquid) particles from non---spherical ice particles?

That is a good suggestion, but no, we did not look at LDR signatures in the cross-pol spectra measurements. Added this suggestion to text (See page 9, line 32 in the Track Change version.)

---

## Author Comment (AC6) · 2 Aug 2018

The comment was uploaded in the form of a supplement:
https://www.atmos-meas-tech-discuss.net/amt-2018-66/amt-2018-66-AC6-supplement.pdf